# An Evaluation of the Antibacterial, Antileishmanial, and Cytotoxic Potential of the Secondary Metabolites of *Streptomyces* sp. ARH (A3)

**DOI:** 10.3390/microorganisms12030476

**Published:** 2024-02-27

**Authors:** Virlanna Larissa Santos de Azevedo, Fernanda Costa Rosa, Leo Ruben Lopes Dias, Lucas Abrantes Batista, Mariana Costa Melo, Luis Alfredo Torres Sales, Abia de Jesus Martins Branco, Thalison Rômulo Rocha Araújo, Rita de Cássia Mendonça de Miranda, Amanda Silva dos Santos Aliança

**Affiliations:** 1Postgraduate Department, Postgraduate Program in Bioscience Applied to Health, Ceuma University, São Luis 65075-120, MA, Brazil; virlanna100159@ceuma.com.br (V.L.S.d.A.); leorubendias@gmail.com (L.R.L.D.); amanda.alianca@ceuma.br (A.S.d.S.A.); 2Postgraduate Department Postgraduate Program in Bionorte, Ceuma University, São Luís 65075-120, MA, Brazil; nandacosttarosa@gmail.com; 3Postgraduate Department, Postgraduate Program in Environment, Ceuma University, São Luis 65075-120, MA, Brazil; 4Postgraduate Program in Parasitic Biology, Universidade Federal do Rio Grande do Norte, Natal 59078-900, RN, Brazil; lucasabrantesbatista@hotmail.com; 5Graduate Department, Ceuma University, São Luis 65075-120, MA, Brazil; marianacmelo@outlook.com.br (M.C.M.); abia.martins4@gmail.com (A.d.J.M.B.); romulorraraujo@hotmail.com (T.R.R.A.); 6Department of Microbiology, PhD in Biological Sciences, University of São Paulo, São Paulo 05508-900, SP, Brazil; luiz79039@ceuma.com.br

**Keywords:** biotechnology, actinomycetes, antimicrobials, leishmanicidal, cytotoxicity

## Abstract

This study aimed to evaluate the antibacterial, leishmanicidal, and cytotoxic potential of metabolites produced by bacteria isolated from rhizosphere soil samples. The bacterium was identified by genome sequencing as *Streptomyces kronopolitis*. A preliminary screening was carried out for the antimicrobial activity of *S. kronopolitis*, demonstrating activity against *Staphylococcus aureus* ATCC 6538, *Corynebacterium diphtheriae* ATCC 27010, *C. diphtheriae* ATCC 27012, and *Mycobacterium abscessus*, with inhibition halos of sizes 25, 36, 29, and 33 mm, respectively. To obtain secondary metabolites, the bacteria were subjected to submerged fermentation, and the metabolites were extracted using the liquid–liquid method with ethyl acetate. There was a similar MIC for *M. abscessus* and the two strains of *C. diphtherium*, reaching a concentration of 12.5 µg/mL, while that of *S. aureus* was 0.048 µg/mL. Assays for leishmanicidal activity and cytotoxicity against HEp-2 cells and red blood cells were performed. The metabolite showed an IC_50_ of 9.0 ± 0.9 µg/mL and CC_50_ of 221.2 ± 7.0 µg/mL. This metabolite does not have hemolytic activity and is more selective for parasites than for mammalian cells, with a selectivity index of 24.6. Thus, the studied metabolite may be a strong candidate for the development of less toxic drugs to treat diseases caused by pathogens.

## 1. Introduction

The activity of microorganisms is important for the broad functioning of soil with regard to the ecosystem services it provides [1,2]. Knowledge of the diversity and function of the microbiome, identification of properties, and assessment of toxic effects resulting from the possible activities of these microorganisms are important [3]. Microorganisms isolated from soil have aroused interest as sources of bioproducts targeted by pharmaceutical industries for the production of new drugs that have minimal effects over time [4].

Among the microorganisms that produce bioactive compounds, those of the Actinomycetales family stand out, especially species of the genus *Streptomyces*, which are known for their ability to produce substances with diverse antimicrobial, leishmanicidal, antimalarial, and antitumor activities [5,6,7,8]. Several authors have focused on isolating bacterial species of the genus *Streptomyces* because of their broad metabolic apparatus. Amorim et al. [9] reported the activity of *S. ansochromogenes* (PB3) against *Pseudomonas aeruginosa* ATCC 15692. More recently, Costa Rosa et al. [10] isolated a *Streptomyces* sp. *Aloe vera* leaf endophyte with activity against *Corynebacerium propinquum* ATCC and a clinical isolate. The antiprotozoal activity of a metabolite produced by the bacteria *Streptomyces* spp. has also been reported. Pagmadulam et al. [11] reported the antiprotozoal activity of four species of *Streptomyces* spp. isolated from soil in Mongolia. Although there are reports of metabolite activity from *Streptomyces* spp., no reports of metabolites produced by the bacterium *Streptomyces kronopolitis* have been published. The advancement of the dissemination of multidrug-resistant bacterial strains has become a worldwide reality and is considered a serious public health problem as it affects not only the hospital community but also the community environment [12]. Reports of multidrug-resistant bacteria have been associated with contaminated environments such as urban rivers [13] and the use of antibiotics in agricultural practices [14]. In addition, within the context of a public health problem, leishmaniasis is neglected, occurring in underdeveloped countries with more vulnerable populations, thereby restricting access to quality health services [15]. Anthropozoonosis is a disease that affects approximately 88 countries and is endemic to Brazil [16]. It has been pointed out in the literature that the use of drugs administered against Leishmania species is questionable in view of their high cost, forms of application, high toxicity, and side effects [17,18].

The main objective of this study was to evaluate the action of antibiotics and leishmanicidal and the cytotoxic activities of microbial metabolites extracted from *Streptomyces* sp.

## 2. Materials and Methods

### 2.1. Soil Collection, Isolation, and Classical Identification of the Microorganism

Rhizospheric soil samples (10 cm deep) were randomly collected at ten different points in a medicinal garden located at a university in the urban area of São Luís, MA, according to the geographical coordinates 3°31′59.6″ S 43°55′55.7″ W.

The ten samples were homogenized to form a composite sample, and the isolation of microorganisms from the rhizosphere was performed according to Clark’s method [19] using Sabouraud agar (BSA) and malt extract agar (EMA) culture media. The experiments were performed in triplicates. The plates were incubated at 30 °C for a maximum of 10 days. Subsequently, each plate was screened, and the macromorphological characteristics of the colonies were observed for purification and transferred to test tubes. Colonies were stored in a refrigerator at 4 °C until further analysis.

The microculture technique was used for classic identification of the isolate. Previously isolated and purified microorganisms were inoculated into Petri dishes containing potato dextrose agar (PDA) medium. A coverslip was partially inserted into the medium to facilitate hyphal growth. The plate was incubated in an incubator at 28 °C for 5 days [20]. Structures such as conidiospores, hyphae, spore chains, and conidia were stained with cotton blue and observed under an optical microscope at 400× magnification. Identification at the genus level was possible by observing macroscopic characteristics and microscopic morphological structures. Criteria recommended by Rapper and Fennell [21], Pitt [22], Samson and Pitt [23], and Klich and Pitt [24] were adopted.

### 2.2. Molecular Identification of the Active Isolate

Molecular identification of the isolated bacteria was performed by DNA amplification and genomic sequencing of the 16S region of bacterial ribosomal DNA (rDNA). DNA extraction was performed using a direct extraction kit (FastDNA^®^ SPIN Kit for Soil) from BIO-101 (MP Biomedicas, Irvine, CA, USA). The purity of the obtained DNA was evaluated by 1% agarose gel electrophoresis (*w*/*v*). DNA samples (5 µL) were mixed with 5 µL of electrophoresis dye and applied to the gels. The gels were subjected to an electric current of 90 V in 0.5× Tris-borate EDTA (TBE) buffer for 1.5 h, stained with ethidium bromide, and photographed under UV light by an image capture system (IMAGO, B&L Systems, Maarssen, The Netherlands).

#### 2.2.1. Genome Sequencing

The genome was sequenced in Botucatu, São Paulo, by Biotecnologia Pesquisa e Inovação (BPI). For the genome sequencing, the Nextera XT Sample Preparation Library Preparation Kit was used, and the genome was analyzed using Illumina NextSeq500 equipment (Illumina, San Diego, CA, USA). The gene sequence of the soil isolate was analyzed using the Basic Local Alignment Search Tool (BLAST) platform to identify similar species. Using the BLAST platform, five species with gene sequences that most resembled the soil isolate were selected. The sequence was submitted to GENBANK/NCBI and deposited under code GCA_014646275.1.

#### 2.2.2. Phylogenetic Tree

For the phylogenetic tree, the species verified by BLAST were compared with species of the genus belonging to the phylum Actinomicetota to determine the degree of ancestry. The sequences were obtained from GenBank, and the genera chosen were Nocardia (*Nocardia asteroides*), Corynebacterium (*Corynebacterium diphtheriae*), Mycobacterium (*Mycobacterium tuberculosis*), and the main antibiotic-producing species of Streptomyces (*S. aureofaciens*, *S. coelicolor*, *S. rapammycinicus*, *S. sviceus*, *S. avermitilis*, *S. griseus*, *S. rimosus*, and *S. albus*).

A phylogenetic tree was constructed using Molecular Evolutionary Genetics Analysis software (MEGA × 10.1). The ClustalW alignment method was used, and neighbor-joining tree method with 1000 bootstrap replicates was used for phylogeny analysis. In addition, the 2-parameter Kimura method was used. The data of the nucleotide sequences encoding proteins were analyzed using the methods above.

### 2.3. Submerged Fermentation

To obtain the active metabolites produced by the isolate, submerged fermentation was carried out in Erlenmeyer flasks (250 mL) containing 50 mL of useful volume of potato dextrose broth (BD). The flasks were incubated with agitation (180 rpm) at 30 °C for five days. Subsequently, the sample was filtered to evaluate its biological activity [6].

#### Liquid–Liquid Extraction

To obtain the metabolites of interest, liquid–liquid extraction was performed using 25 mL of ethyl acetate and 25 mL of cell-free extract in a separation funnel, shaking vigorously for 15 min and waiting for it to decant for another 15 min [25]. The microbial cells were separated. The fermented medium was subjected to centrifugation and filtered again through a 22 μm filter, ensuring that the extract did not contain microbial cells.

The extraction of the compounds of interest, called Streptomyces metabolites (MS), was performed using 25 mL of the filtrate, and 25 mL of ethyl acetate was added to a separating funnel, vigorously stirred for 10 min, and left to rest for 30 min. Subsequently, the organic phases containing the analytes of interest were collected. The solvent was evaporated using a rotary evaporator, and the product yield was determined.

To evaluate the biological activities, the MS extract was resuspended in dimethyl sulfoxide (DMSO) to a known concentration of 1000 μg/mL.

### 2.4. Screening of Antibacterial Tests

#### 2.4.1. Microorganisms Used

For the evaluation of the antimicrobial potential of bacterial isolates, the following microorganisms were used: *Staphylococcus aureus*—ATCC 6538; *C. diphtheriae*—ATCC27010; *C. diphtheriae*—ATCC 27012; and *Mycobacterium abscesses* (IC). The bacteria used were selected for their ability to cause serious diseases such as endocarditis and tuberculosis in immunocompromised patients, in addition to having multiresistant strains. The bacterial strains belong to the culture collection of the Biotechnology and Electrochemistry Laboratory at Ceuma University.

To evaluate leishmanicidal potential, promastigotes of *Leishmania amazonensis* (MHOM/BR/76/MA-76) were used. Promastigotes were maintained at 26 °C in Schneider’s medium (Sigma, Darmstadt, Germany) supplemented with 10% fetal bovine serum, with replications every three days. Parasites in the exponential growth phase were used for experiments. Leishmania strains belong to the culture collection of the Cellular and Molecular Biology Laboratory, Parasitology Department, FIOCRUZ/PE.

#### 2.4.2. Test in Solid Medium

The antimicrobial activity test was carried out in a solid Mueller–Hinton (MH) medium (Sigma, Darmstadt, Germany) through diffusion of the bioactive compound in agar, using the method described by Uchida et al. [26]. After ten days of incubation at 28 °C, the circular agar blocks of 6 mm in diameter were removed from the plates with colonies cultivated by plugging and transferring to the plates containing Mueller–Hinton medium, previously seeded with the standard microorganisms in 5 × 10^5^ colony-forming unit (CFU)/mL. This test was performed in triplicate. After culturing for 24 h at a temperature of 37 °C, the diameters (mm) of the inhibition halos of each block were measured by measuring the greatest distance between 2 straight points that crossed the glucose block in the middle. The Matsuura scale [27] was used to classify the results and obtain the arithmetic mean and standard deviation. The results were expressed as mean zones of inhibition diameter (IDZ) in millimeters (mm). Means were compared by Tukey’s statistical test using Prism3.0 software; they were considered statistically significant when *p* < 0.05.

#### 2.4.3. Agar Diffusion Assay

The liquid medium assay was performed using the plate diffusion test [28] to determine whether the strain ARH (A3) secreted metabolites into the external environment. The plaque diffusion test was established as a standard by the Clinical & Laboratory Standards Institute (CLSI) [29]. Fermented must of the previous assay (10 µL) was applied to wells of 6 mm in diameter made in Petri dishes with 20 mL of Mueller–Hinton agar medium; they were seeded with the pathogenic microorganisms and incubated at 37 °C for up to 72 h. Subsequently, the IDZs were measured using tweezers. In the 6 mm wells, 10 μL of DMSO and 10 μL of chloramphenicol (30 μg) were used as the negative and positive controls, respectively. The test was performed in triplicates to calculate the mean and standard deviation. Means were compared by Tukey’s statistical test using Prism3.0 software; they were considered statistically significant when *p* < 0.05.

#### 2.4.4. Determination of Minimum Inhibitory Concentration (MIC)

To determine the minimum concentration capable of inhibiting bacterial growth, a microdilution test was performed using a 96 mm multi-well plate, and MS was diluted in DMSO. The calculation used to determine the concentration followed the CLSI protocol [18], which recommends 1000 μg/mL.

The technique described by Zogda and Porter [30] was used in this study. Mueller–Hinton broth (190 μL) and MS (10 μL) diluted in DMSO at an initial concentration of 1000 μg/mL were dispensed into the first row of wells. In the other wells, 100 μL of Mueller–Hinton broth medium was added. Then, serial dilutions were performed in nine consecutive wells, removing 100 μL.

The highest concentration well resulted in dilutions from 100 μg/mL to 0.0625 μg/mL. The inoculum and extract were not added to the penultimate well as a negative control. In the last well, the medium and inoculum were added as positive controls. Microbial growth was determined based on the growth of bacterial colonies. The concentration of the bacterial suspension was determined to be 5 × 10^5^ CFU/mL according to the McFarland scale. The plate was incubated at 37 °C for 24–48 h. Bacterial viability was evaluated by adding 30 µL of resazurin after 24 h of incubation. Wells in which resazurin remained blue were read as inhibiting microbial growth, and in those where resazurin changed to pink color, the extract did not inhibit the growth of the microorganism.

### 2.5. In Vitro leishmanicidal Activity

To determine the leishmanicidal activity, promastigotes of *Leishmania amazonensis* were incubated in the presence of increasing concentrations of the MS metabolite (6.25 to 100 μg/mL). The parasites were diluted to a concentration of 1 × 10^6^ parasites/mL. Promastigotes incubated only with Schneider’s medium were used as controls. After 72 h of incubation, surviving parasites were counted in a Neubauer chamber (iNCYTO CChip DHC-N01; Cheonan-Si, Republic of Korea). The IC^50^ value (concentration that inhibits parasite growth by 50%) was determined by linear regression analysis using SPSS software (version 8.0; IBM Co., New York, NY, USA) for Windows. Each experiment was performed in duplicate and technical triplicate [31].

### 2.6. Cytotoxic Action of Metabolites in Mammalian Cells

Cytotoxicity analyses were performed using the MTT technique (3-(4,5-dimethylthiazol-2-yl)-2,5-diphenyl tetrazolium bromide); 100 µL HEp-2 cell suspension at a concentration of 1 × 10^4^ cells/mL was used. The cells were arranged in 96-well plates under five MS metabolite concentrations (12.5 to 200 µg/mL), except for controls that were filled with culture medium and incubated in an oven at 37 °C and 5% CO_2_ for 24 h. After incubation, MTT was added to the plate wells and incubated again for 3 h in an incubator at 37 °C and 5% CO_2_ in the dark. DMSO was added before the reading to solubilize the formazan crystals that were solidified by MTT reduction. The plates were analyzed by spectrophotometry at 540 nm using an enzyme-linked immunosorbent assay (ELISA) reader [32].

Finally, the percentage of viable cells was calculated; from this result, it was possible to calculate the percentage of non-viable cells. The concentration capable of causing a cytotoxic effect in 50% of the cells (CC_50_) was estimated through logarithmic regression analysis of the data obtained using SPSS 8.0 software for Windows. The selectivity index (SI) was determined using the ratio of CC_50_ and IC_50_ values. Each experiment was performed in two separate experiments in duplicates.

### 2.7. Evaluation of the Hemolysis Index

The hemolysis index of the metabolites was determined according to the method described by Wang et al. [33]. A 1% commercial sheep blood solution was prepared using 2% phosphate-buffered saline (PBS) and a working solution of the compounds. After preparing the solutions, tests were set up in a 96-well plate with 5 MS concentrations (6.25 to 100 µg/mL), a positive control with oxygen peroxide (H_2_O_2_), and a negative control with saline. Each well was composed of 100 µL of red blood cell solution plus 100 µL of working solution, except for controls, and incubated for 3 h under agitation at 37 °C. Finally, the absorbance was measured spectrophotometrically (540 nm), and the hemolytic activity values were calculated using the following formula:(1)$$Hemolysis(\%)=\frac{(Treated\;Abs.-Negative\;control\;Abs.)\times 100}{\left(Positive\;control\;Abs.-Negative\;control\;Abs.\right)}$$

## 3. Results

### 3.1. Isolation and Classical Identification of Microorganisms

The strain ARH (A3) is visualized in Figure 1A, where the growth of a typical colony of actinomycetes is observed; it is round, with limited borders and the presence of grayish aerial mycelia, forming an inhibition zone around the colony that already produces and secretes a bioactive compound as indicated by the arrow.

Macro- and micromorphological observations of colonies in a Petri dish and under a microscope, respectively, predict the classic identification of a microorganism. In this way, some macromorphological characteristics were considered by analyzing the culture in a Petri dish, such as the presentation of a colony that starts with the appearance of white mycelium and ends with a grayish color containing aerial mycelium releasing a darkened pigment, as shown in Figure 1B. The micromorphological analysis, however, showed structures such as gray chains of spores, as shown in Figure 1C, and micellar ramifications resembling hyphae of filamentous fungi, where such characteristics are compatible with actinobacteria belonging to the genus *Streptomyces* as indicated by the arrow, as shown in Figure 1D.

### 3.2. Molecular Identification

#### 3.2.1. Genome and 16S rRNA Gene

As per genome sequencing analysis, the lineage of the the strain ARH (A3) belongs to the phylum. Actinomycetoma, family Streptomycetaceae, genus *Streptomyces*, and species *Streptomyces* sp. The genome contains 9,022,973 bp of linear DNA with 79 RNA sequences, 8540 protein-coding genes (PEGs), and high G + C content, with a 16S rRNA gene of 1531 bp. The BLAST platform analysis showed five sequences that most resembled ARH isolate (A3), which were *S. kronopolitis*, *S. chattanoogensis, S. lydicus, S. nigrescens*, and *S. chrestomyceticus*. The closest species was *S. kronopolitis* strain NEAU-ML8, with 99.77% identity and a maximum score of 2750 (see Table 1).

#### 3.2.2. Phylogenetic Tree

The phylogenetic tree shows the degree of relatedness between *Streptomyces* sp. ARH(A3) gene sequences with those of the five closest species in GenBank and species from the phylum Actinomycetoma. The percentage of data coverage for the internal nodes is shown in Figure 2. The optimal tree with the sum of the verified branch lengths was 0.34456757.

### 3.3. Antibacterial Test Screening

#### 3.3.1. Test in Solid Medium

The strain ARH (A3) was tested against pathogenic bacteria of clinical interest, and the results showed that actinomycetes presented satisfactory results against all tested bacteria, forming inhibition halos for *S. aureus* ATCC 6538, *C. diphtheriae* ATCC 27010, *C. diphtheriae* ATCC 27012, and *M. abscessus* (IC) at 25, 36, 29, and 33 mm, respectively, as shown in Table 2.

#### 3.3.2. Agar Diffusion Assay

To evaluate the potential of the active metabolite secreted by *S. kronopolitis*, agar diffusion was performed in a liquid medium, followed by liquid–liquid extraction using ethyl acetate. The crude extract was obtained with a yield of 72.6 mg g/mL and tested against the same clinical pathogens as in the aforementioned assay; satisfactory results were obtained, showing the ability of *S. kronopolitis* to secrete active metabolites into the extracellular environment (see Table 3).

#### 3.3.3. Determination of Minimum Inhibitory Concentration (MIC)

After the production and extraction of secondary metabolites from the strain ARH (A3), the MICs were evaluated according to the CLSI protocol [18]. Calculations were performed using the initial concentration of the extract at 100 µg/mL for the MIC test. The MIC end point was the lowest concentration of each extract in which there was no color change; it was observed that the MS of *M. abscessus* and the two strains of *C. diphtherium* were similar, reaching a concentration of 12.5 µg/mL; in *S. aureus*, the minimum concentration was 0.048 µg/mL as shown in Table 4.

#### 3.3.4. In Vitro leishmanicidal Activity

The data shown in Figure 3 demonstrate that the metabolite tested against promastigote forms of *L. amazonensis* was able to inhibit growth from the lowest concentration of the metabolite (6.25 µg/mL). As shown in Figure 3, 100% inhibition of parasite growth was observed at the three highest concentrations tested. The IC_50_/72 h value was calculated for MS and showed a value of 9.0 ± 0.9 µg/mL.

### 3.4. Cytotoxic Action of the Metabolite in Mammalian Cells

Figure 4 shows that the MS presented toxicity at the highest concentration (200 µg/mL) tested for this metabolite. It is also possible to note that in the other concentrations, cell viability was greater than 85%, which indicates low toxicity to mammalian cells. Concentrations that resulted in cell viability greater than 70% were considered non-toxic, as reported by Tunes [34].

The CC_50_ values of the metabolites were calculated to be 221.2 ± 7.0 µg/mL. The selectivity index (SI) (CC_50_/IC_50_), which indicates the selectivity of the compound for the parasite in relation to mammalian cells, was calculated. This metabolite had a selectivity index (SI) of 24.6.

### 3.5. Evaluation of the Hemolysis Index

The toxic effect on erythrocytes is due to the interaction of substances with the cell membrane, mainly through sterols such as cholesterol, which leads to a deformity in the erythrocyte membrane, resulting in extravasation [16,35].

The DM hemolysis index was determined, as shown in Figure 5. At all extract concentrations, the values were very close to those of the control red blood cells immersed in PBS (negative control), with no statistical discrepancy. Therefore, the metabolites did not exhibit hemolytic activity at the tested concentrations.

## 4. Discussion

Soil is an abundant and diverse system of microorganisms and, therefore, is an excellent source for microbial prospecting. In this study, a bacterium with a characteristic actinomycete colony was isolated from the soil. Several authors have published papers on prospecting for this bacterial group in the soil [2,9,36]. Micromorphological identification revealed the presence of spiral spores and arthrospores, typical for *Streptomyces* bacteria. The same characteristics were observed by Al Dhabi et al. [2], Amorim et al. [6], Costa Rosa et al. [10], Ensign [37], and Sholkamy et al. [38], who isolated bacteria of the genus *Streptomyces* from environmental samples.

Identification of promising species is essential for understanding and cataloging the potential of this microbial group. Bacteria of the genus *Streptomyces* have been widely reported as important producers of metabolites of biotechnological interest, and the search for new species is of paramount importance for the biotechnology industry. The molecular identification analysis determined that it was a bacterium from the actinomycetes group with the closest degree of similarity to the species *S. kronopolitis*. Other authors have identified environmental isolates through similar molecular techniques. Saraswathi et al. [36] isolated species of actinomycetes from rhizosphere soil and, after screening bacteria with biotechnological potential, used molecular techniques to identify *S. cangkrigensis*. Pagmadulan et al. [9] isolated four species, *S. canus*, *S.s cirratus*, *S.s bacillaris*, and *S. peucetius*, from the soil in Mongolia and identified them using the ribosomal 16S technique.

Bacteria of the genus *Streptomyces* sp. are known for their ability to produce compounds of biotechnological interest, such as enzymes and antibiotics [2,9]. In this study, the bacterium *Streptomyces* sp. ARH (A3) showed activity against different bacteria of clinical interest in tests in solid medium and submerged fermentation, demonstrating its ability to produce and secrete active compounds into the external environment. Abba et al. [39] pointed out that its compounds, after submerged fermentation, have antimicrobial action against Gram-positive bacteria with 20 mm inhibition halos for *Bacillus subtilis.* The secreted compound showed activity against *S. aureus* (ATCC 6538), *C. diphtheriae* (27012), *C. diphtheriae* (27010), and *M. abscesses*, with inhibition halos larger than 20 mm. The isolation of bacteria with activity against isolates of clinical interest is of paramount importance because of the increasingly frequent appearance of multidrug-resistant strains [12].

The MIC of the tested metabolites was established for the sensitive bacteria, with *S. aureus* having the lowest MIC. Amorim et al. [6] reported that the MIC for *P. aeruginosa* was 0.5 mg/mL using metabolites from the same microorganism (*Streptomyces* spp.), even though the pathogen was Gram-negative. However, Huang et al. [40] tested purified metabolites from microorganisms against Gram-positive bacteria and obtained an MIC_50_ of 12.5 µg/mL. Potentially, because MS is not yet purified, it requires a higher concentration to reach lethality; therefore, it shows promise for complementary future tests in its purified form.

The MS has a significant inhibitory effect on *L. amazonensis*. Although bacteria of the genus *Streptomyces* are recognized as important sources of secondary metabolites with diverse biological activities [6,7,8,25,26,41], there are no reports on the biological action of *S. kronopolitis* against the genus Leishmania. The leishmanicidal activity of *Streptomyces* sp. has been reported in several studies, such as that by Sreedharan and Rao [42] and Aliança et al. [43], who analyzed the effect of a potential biological compound inhibitor against *Leishmania donovani*. This study also corroborates the work of Amorin et al. [6], who evaluated the bactericidal and leishmanicidal potential of *S. ansochromogenes*. 

In this study, the low toxicity of the studied metabolite was also demonstrated, which corroborates the results of studies by Aliança [31], where macroalgal extracts were more selective to parasites than to mammalian cells (IS > 1), and those of the studies by Trombini [44], which stated that metabolites can be cytotoxic and capable of affecting both cell proliferation and the immune system. In the hemolysis assay, low hemolytic activity was observed, demonstrating the low toxicity of the metabolite. Szabo [45] conducted hemolysis tests with secondary metabolites of herbal medicines and concluded that there was no risk of hemolysis at concentrations where the index was equal to or lower than that of the negative control.

In this study, the antimicrobial potential of an extract resulting from the secondary metabolism of *Streptomyces* sp. was demonstrated. Secondary metabolites are a mixture of compounds produced at the end of the exponential phase of microorganisms and are essentially associated with their survival in the environment, which makes these compounds interesting for pharmaceutical industries. Of the several chemical classes that make up crude extracts, reports in the literature show that terpenes, flavonoids, and alkaloids are the majority when it comes to metabolites isolated from actinobacteria [46].

Actinobacteria have a great potential to produce secondary metabolites. Between the years 2017 and 2021, 589 manuscripts were published describing new compounds from different chemical classes isolated from actinobacteria, of which 52% demonstrate one or more biological activities. The most predominant classes were macrolides, quinolones, and small peptides with antimicrobial and cytotoxic potential. It is important to say that of the 589 new compounds, 69% were produced by Streptomyces, demonstrating the biotechnological importance of the genus [47].

The mechanisms of action of some of these compounds are reported in the literature. Antimicrobials generally act on the microorganism’s cell wall, disorganizing the peidioglycan chain (vancomycin); on the plasma membrane, depolarizing the membrane (daptomycin); on protein synthesis, blocking peptide transferase (chloramphenicol); and on RNA synthesis, inhibiting DNA—dependent RNA polymerase (rifamycin) [47].

## 5. Conclusions

The filamentous bacterium *Streptomyces* sp. ARH (A3) is potentially promising because it contains active metabolites that can be extracted from bacterial cells, with antibacterial biological activity against Gram-positive bacteria and antiparasitic action against *L. amazonensis* in the promastigote form. Furthermore, the extracted MS showed good viability after the MIC results, also presenting a non-toxic action on cells when tested at low concentrations; thus, it is a strong candidate for the search for new bioactive compounds.

Studies are being carried out to isolate and identify the target molecule, considering that the crude extract was promising for different organisms, suggesting that yet another molecule is active.

## Figures and Tables

**Figure 1 microorganisms-12-00476-f001:**
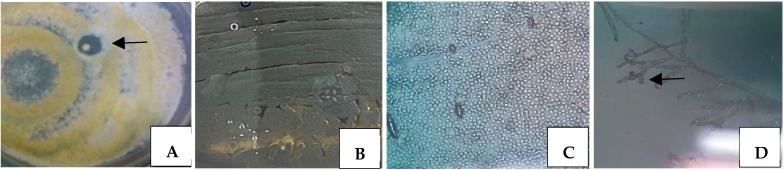
Colony isolation with macromorphological appearance of actinomycetes with formation of an inhibition zone showing production of bioactive compound as indicated by the arrow (**A**), macromorphological characteristics of the colony with aerial mycelium showing grayish pigmentation (**B**), and micromorphological forms of the same colony showing isolated spores (**C**) (400×) and arranged in chain as indicated by the arrow (**D**) (400×).

**Figure 2 microorganisms-12-00476-f002:**
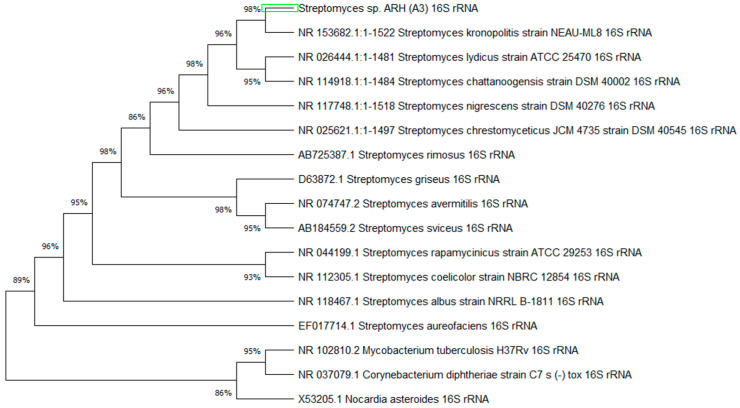
Neighbor-joining tree based on 16S rRNA gene sequences of *Streptomyces* sp. ARH(A3) 16S rRNA gene. Evolutionary history was inferred by comparing species from the BLAST analysis and genera *Streptomyces, Mycobacterium*, *Corynebacterium*, and *Nocardia* to rRNA-related strains of bacteria of the genus *Streptomyces*.

**Figure 3 microorganisms-12-00476-f003:**
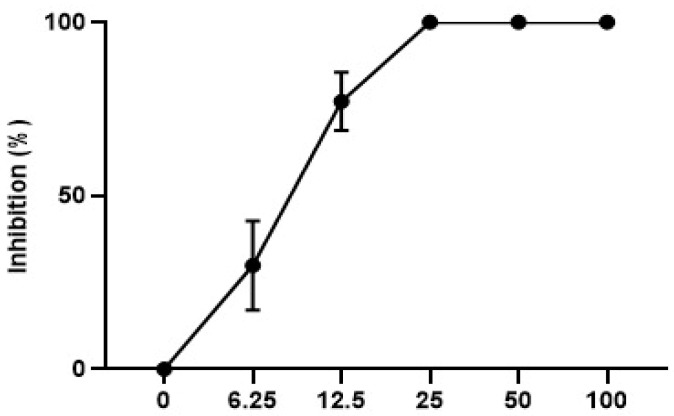
Growth inhibition curve of *Leishmania amazonensis* promastigotes treated with metabolite of *Streptomyces* sp. ARH (A3).

**Figure 4 microorganisms-12-00476-f004:**
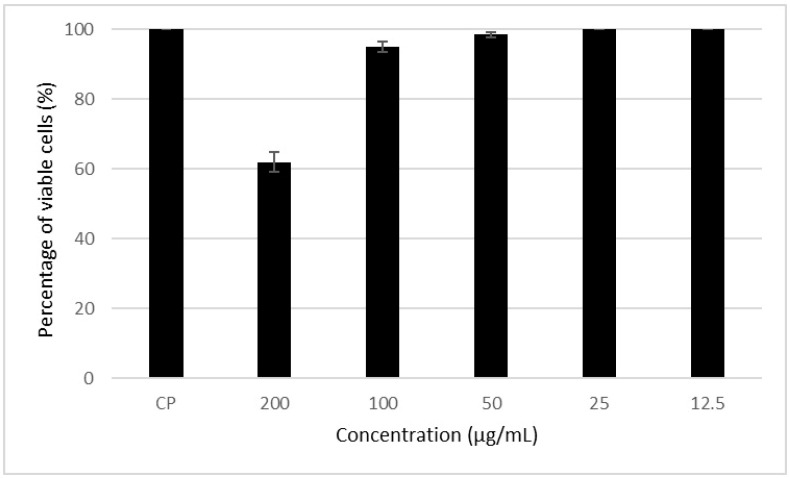
Cytotoxic action of the metabolite (MS) of *Streptomyces* sp. ARH (A3) in mammalian cells (HEp-2 cells).

**Figure 5 microorganisms-12-00476-f005:**
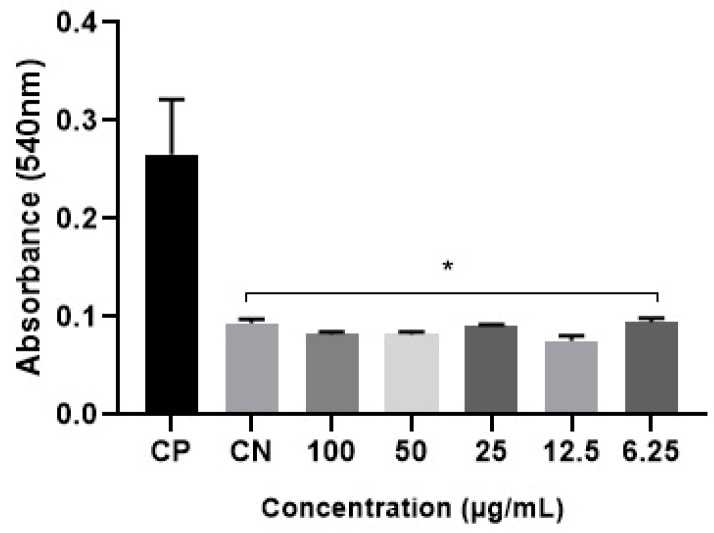
Evaluation of the SM hemolysis index. Columns marked with an asterisk (*) represent the statistically significant difference (* = *p* ≤ 0.01) in relation to the positive control group (PC) by the ANOVA test (*p* < 0.05) and post-Dunnett’s test.

**Table 1 microorganisms-12-00476-t001:** BLAST of the species most similar to the soil isolate strain ARH (A3). Description: the name of the species; max score: the highest alignment score; identity percentage: number that describes the similarity of the query sequence with the target sequence; membership: sequence access at NCBI.

Description	Max Score	Percent Identity	Accession
*Streptomyces kronopolitis* strain NEAU-ML8.^T^	2750	99.77%	NR_153682.1
*Streptomyces nigrescens* strain DSM 40276	2734	99.20%	NR_117748.1
*Streptomyces lydicus* strain ATCC 25470	2715	99.60%	NR_026444.1
*Streptomyces chattanoogensis* strain DSM 40002	2712	99.28%	NR_114918.1
*Streptomyces chrestomyceticus* JCM 4735 strain DSM 40545	2706	99.14%	NR_025621.1

**Table 2 microorganisms-12-00476-t002:** Diameters of the inhibition halos (mean and SD) formed by strain ARH (A3) in millimeters against the clinical pathogens tested in the solid medium assay. Negative control with DMSO did not show a zone of inhibition. Positive control with chloramphenicol for inhibition zones > 30 mm.

Clinical Pathogens	Inhibition Halos (mm)
*Staphylococcus aureus* (ATCC 6538)	25.3 ^b^ ± 6.8
*Corynebacterium diphtheriae* (27012)	29 ^b^ ± 1
*Corynebacterium diphteriae* (27010)	36 ^a^ ± 5.6
*Mycobacterium abscessus*	33 ^a^ ± 3

^a,b^—same letter—does not present statistical significance; different letters—presents statistical significance.

**Table 3 microorganisms-12-00476-t003:** Diameters (mean and SD) of inhibition halos in mm formed by strain ARH (A3) against the clinical pathogens tested in the agar diffusion assay. Negative control with DMSO did not show a zone of inhibition. Positive control with chloramphenicol for inhibition zones > 30 mm.

Clinical Pathogens	Inhibition Halos (mm)
*Staphylococcus aureus* (ATCC 6538)	20.3 ^b^ ± 1.5
*Corynebacterium diphtheriae* (27012)	20.6 ^b^ ± 1.1
*Corynebacterium diphteriae* (27010)	28.6 ^a^ ± 0.57
*Mycobacterium abscessus*	30.6 ^a^ ± 3

^a,b^—same letter—does not present statistical significance; different letters—presents statistical significance.

**Table 4 microorganisms-12-00476-t004:** MS Minimum Inhibitory Concentration (MIC) against previously selected pathogens.

Metabolites	Clinical Pathogens			
	*M. abscessos* (IC)	*S. aureus* ATCC 6538	*C. diphterium* ATCC 27010	*C. diphterium* ATCC 27012
*Streptomyces* sp. ARH (A3)	12.5 µg/mL	0.048 µg/mL	12.5 µg/mL	12.5 µg/mL

## Data Availability

The datasets used and/or analyzed during the current study are available from the corresponding author upon reasonable request. The datasets generated and/or analyzed during the genome sequencing are available in the NCBI repository; for datasets, download genome accession https://www.ncbi.nlm.nih.gov/datasets/taxonomy/1612435/, accessed on 20 October 2023 RefSeq assembly GCF_014646275.1 GCF_014646275.1—include gff3, rna, cds, protein, genome, seq-report—filename GCF_014646275.1.zip and Submitted GenBank assembly GCA_014646275.1.

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
