# Peer review of "An Evaluation of the Antibacterial, Antileishmanial, and Cytotoxic Potential of the Secondary Metabolites of Streptomyces sp. ARH (A3)"

_microorganisms, 2024, doi:10.3390/microorganisms12030476_

Round 1

Reviewer 1 Report

Comments and Suggestions for Authors

1. The defining thresholds for prokaryotic microbial species included 16S rRNA gene similarity > 98.65%, DNA-DNA hybridization (DDH) value 70%, and Average Nucleotide Identity(ANI) value 95%. Even if the 16S rRNA gene similarity of two strains is equal to 100%, their DNA-DNA may be less than 70% or ANI value less than 95%. Therefore, the similarity of 16S rRNA gene between the strain Streptomyces sp. ARH (A3) isolated by the authors and Streptomyces kronopolitis NEAU ML8T, which was the closest to it, is 99.97%. It doesn't mean the strain ARH (A3) belongs to Streptomyces kronopolitis. Therefore, it is suggested that the author change the title of the article to ” Evaluation of the Antibacterial, Antileishmanial,and Cytotoxic Potential of Secondary Metabolites of Streptomyces sp. ARH (A3)”.

2. Line 102: The “ASM1464627v1” does not appear to be an Accession Number deposited in GenBank. Please provide a correct Accession Number.

3. In Figure 1, the author should use arrows to mark the features he wants to show, and the magnification of C and D was also not indicated. In addition, the pictures are not clear and the features are not typical, in particular, the spiral spore chain described by the authors was not obvious. Advised to provide clear photos again.

4. In table 1, the name of the species should be written normatively, for example, the first line should be written as Streptomyces kronopolitis NEAU-ML8T. T stands for type strain and should be superscripted when written.

5. In figure 2, the names of species in the branch should also be written simplistically and normatively. for example, Streptomyces kronopolitis NEAU-ML8T (NR 153682). In addition, the strains used in the construction of trees should be type strains, and the strain number and the superscript T should be marked.

6. There were some confusing descriptions throughout the article, such as the isolation that was obtained, sometimes described as “the isolation of soil microorganisms”(line 229), sometimes described as “the sample”(line 249), “Streptomyces sp. ARH (A3)”(line 257), “S.kronopolitis”(line274), and “Streptomyces spp.”(line 287 and 289), which should be written uniformly as “the strain ARH (A3)” or “the Streptomyces sp.ARH (A3)”.

7. Some other minor errors were marked in the manuscript.

Author Response

Review 1.

The authors are grateful for the suggestions, following the changes made according to the suggestions of the reviewers.

  1. The defining thresholds for prokaryotic microbial species included 16S rRNA gene similarity > 98.65%, DNA-DNA hybridization (DDH) value ≥70%, and Average Nucleotide Identity(ANI) value ≥95%. Even if the 16S rRNA gene similarity of two strains is equal to 100%, their DNA-DNA may be less than 70% or ANI value less than 95%. Therefore, the similarity of 16S rRNA gene between the strain Streptomyces sp. ARH (A3) isolated by the authors and Streptomyces kronopolitis NEAU ML8T, which was the closest to it, is 99.97%. It doesn't mean the strain ARH (A3) belongs to Streptomyces kronopolitis. Therefore, it is suggested that the author change the title of the article to ” Evaluation of the Antibacterial, Antileishmanial,and Cytotoxic Potential of Secondary Metabolites of Streptomyces sp. ARH (A3)”.

Answer. Done as suggested

  1. Line 102: The “ASM1464627v1” does not appear to be an Accession Number deposited in GenBank. Please provide a correct Accession Number.

Answer. Done as suggested

  1. In Figure 1, the author should use arrows to mark the features he wants to show, and the magnification of C and D was also not indicated. In addition, the pictures are not clear and the features are not typical, in particular, the spiral spore chain described by the authors was not obvious. Advised to provide clear photos again.

Answer. Done as suggested

  1. In table 1, the name of the species should be written normatively, for example, the first line should be written as Streptomyces kronopolitis NEAU-ML8T. T stands for type strain and should be superscripted when written.

Answer. Done as suggested

  1. In figure 2, the names of species in the branch should also be written simplistically and normatively. for example, Streptomyces kronopolitis NEAU-ML8T (NR 153682). In addition, the strains used in the construction of trees should be type strains, and the strain number and the superscript T should be marked.

Answer. Done as suggested

  1. There were some confusing descriptions throughout the article, such as the isolation that was obtained, sometimes described as “the isolation of soil microorganisms”(line 229), sometimes described as “the sample”(line 249), “Streptomyces sp. ARH (A3)”(line 257), “S.kronopolitis”(line274), and “Streptomyces spp.”(line 287 and 289), which should be written uniformly as “the strain ARH (A3)” or “the Streptomyces sp.ARH (A3)”.

Answer. Done as suggested

  1. Some other minor errors were marked in the manuscript.

Answer. Done as suggested

P.S. changes are marked in the manuscript in yellow

Reviewer 2 Report

Comments and Suggestions for Authors

The manuscript titled "Evaluation of the Antibacterial, Antileishmanial, and Cytotoxic  Potential of Secondary Metabolites of Streptomyces kronopolitis" showed the antibacterial, cytotoxicity and antileishmanial activity of the crude extract

my advice to run LC/MS/MS to show the secondary metabolites responsible for these activities

suggest the targeted pathway through docking these metabolites against suggested targets using molecular docking and dynamics

Comments on the Quality of English Language

English editing by native English speaker is needed

Author Response

Review 2. The manuscript titled "Evaluation of the Antibacterial, Antileishmanial, and Cytotoxic  Potential of Secondary Metabolites of Streptomyces kronopolitis" showed the antibacterial, cytotoxicity and antileishmanial activity of the crude extr act

The authors are grateful for the suggestions, following the changes made according to the suggestions of the reviewers.

  1. my advice to run LC/MS/MS to show the secondary metabolites responsible for these activities

Answer. Firstly, we would like to thank you for your suggestions and contributions to the manuscript. The authors would like to point out that we saw some limitations in the study, and the identification of the compound was one of them. This extract result was obtained at the beginning of 2022 and at the end of the year we had an idea of its biological potential and would begin studies with the separation using SPE (Solid Phase Extraction) cartridges for further testing and identification of active molecules related to the pandemic. This part of the study that would be carried out at the Analytical Center of the ITP/UNIT Institute of Technology and Research was largely compromised. It was only in the middle of 2023 that the analytical center in which we would have LC MS started working again, reducing this problem.

Currently the molecule is already separated and being used for subsequent chemical identification that will be published in future studies.Currently the molecule is already separated and being used for subsequent chemical identification that will be published in future studies.

  1. suggest the targeted pathway through docking these metabolites against suggested targets using molecular docking and dynamics.

Answer: The authors thank you for the suggestion and guarantee that it will be carried out in future studies.

Reviewer 3 Report

Comments and Suggestions for Authors

The manuscript under title (Evaluation of the Antibacterial, Antileishmanial, and Cytotoxic  Potential of Secondary Metabolites of Streptomyces kronopolitis) investigated the antimicrobial, leichmanicidal, and cytotoxic activity of S. konoples. It is well organized and conducted. There minor corrections and questions:

Molecular Identification of the Active Isolate on which sample; soil sample or cultured bacteria?

 Sources of the tested MOs and Lieshmania isolate should be mentioned

Table 2, 3, 4; where is the control positive?

The title antilieshmanial result was missed

Conclusion needs a sentence about a future studies on this metabolite as chemical structure

Author Response

Review 3. The manuscript under title (Evaluation of the Antibacterial, Antileishmanial, and Cytotoxic  Potential of Secondary Metabolites of Streptomyces kronopolitis) investigated the antimicrobial, leichmanicidal, and cytotoxic activity of S. konoples. It is well organized and conducted. There minor corrections and questions:

The authors are grateful for the suggestions, following the changes made according to the suggestions of the reviewers.

  1. Molecular Identification of the Active Isolate on which sample; soil sample or cultured bacteria?

Answer. The molecular identification was of the isolated bacteria. In item 2.2. Molecular Identification of the Active Isolation of the isolated bacteriae was explained by the authors that the identification was carried out in the isolated bacteria highlighted.

  1. Sources of the tested MOs and Lieshmania isolate should be mentioned

Answer: done as suggested

  1. Table 2, 3, 4; where is the control positive?

Answer: done as suggested

  1. The title antilieshmanial result was missed

Answer: done as suggested

  1. Conclusion needs a sentence about a future studies on this metabolite as chemical structure

Answer: done as suggested

Reviewer 4 Report

Comments and Suggestions for Authors

The Streptomyces metabolites showed considerable inhibition of clinical pathogens while showing low cytotoxicity and low hemolytic activity, suggesting potential for human use.

Major concern is that it is unclear which metabolite (metabolite from which Streptomyces) was tested. Because of the lack of the activity – safety balance, it is hard to discuss the applicability to human use.

The use of crude extract and not using fractions or isolated metabolite limiting the value of the research.

1)     Line 296 Table 4: Streptomyces species from which MS was yielded should be clarified.

2)     Line 303 Figure 3: Streptomyces species from which MS was yielded should be clarified.

3)     Line 316 Figure 4: Streptomyces species from which MS was yielded should be clarified.

4)     Line 124: Concrete procedure for extraction should be described briefly.

Minor point

5)     Line 17 Expression “molecular techniques” is ambiguous and should be concrete, such as genome sequencing.

Non-binding comment

6)     If there are special intention in the Streptomyces screening in this study, it should be expressed in introduction.

Author Response

Review 4. The Streptomyces metabolites showed considerable inhibition of clinical pathogens while showing low cytotoxicity and low hemolytic activity, suggesting potential for human use.

The authors are grateful for the suggestions, following the changes made according to the suggestions of the reviewers.

  1. Line 296 Table 4: Streptomyces species from which MS was yielded should be clarified

Answer. done as suggested

  1. Line 303 Figure 3: Streptomyces species from which MS was yielded should be clarified.

Answer: done as suggested

  1. Line 316 Figure 4: Streptomyces species from which MS was yielded should be clarified.

Answer: done as suggested

  1. Line 124: Concrete procedure for extraction should be described briefly.

Answer: done as suggested

  1. Line 17 Expression “molecular techniques” is ambiguous and should be concrete, such as genome sequencing.

Answer: done as suggested

  1. If there are special intention in the Streptomyces screening in this study, it should be expressed in introduction.

Answer :This interest was expressed in the conclusion as a future study

Round 2

Reviewer 2 Report

Comments and Suggestions for Authors

Thanks for your reply, and I really understand your issues, but to make it much better

1- highlight through literature what are the major metabolites suggested in this crude extract

2- disscuss the metabolites rule related to the

activities

3- disscus which metabolites you suggest related to the activities and what targets they are hitting

Comments on the Quality of English Language

English need more revision

Author Response

Review 1 and 2. Thanks for your reply, and I really understand your issues, but to make it much better:

The authors are grateful for the suggestions, following the changes made according to the suggestions of the reviewers.

1- highlight through literature what are the major metabolites suggested in this crude extract

Answer: done. Suggestions have been added to the discussion and are marked in green

2- disscuss the metabolites rule related to the activities

Answer: done. Suggestions have been added to the discussion and are marked in green

3- disscus which metabolites you suggest related to the activities and what targets they are hitting

Answer: done. Suggestions have been added to the discussion and are marked in green

Reviewer 4 Report

Comments and Suggestions for Authors

The manuscript was adequately revised based on the comments provided in previous review.

Though the value of the research itself is limited as the evaluation was done in crude extract, the manuscript seems to reflect research outcomes and it is of highest quality reasonably achievable.

Author Response

The authors are grateful for the suggestions